# Changes in ImPACT Cognitive Subtest Networks Following Sport-Related Concussion

**DOI:** 10.3390/brainsci13020177

**Published:** 2023-01-20

**Authors:** Grace J. Goodwin, Samantha E. John, Bradley Donohue, Jennifer Keene, Hana C. Kuwabara, Julia E. Maietta, Thomas F. Kinsora, Staci Ross, Daniel N. Allen

**Affiliations:** 1Department of Psychology, University of Nevada, Las Vegas, NV 89154, USA; 2Department of Brain Health, University of Nevada, Las Vegas, NV 89154, USA; 3College of Liberal Arts, University of Nevada, Las Vegas, NV 89154, USA; 4Department of Psychiatry and Behavioral Sciences, University of Oklahoma Health Sciences Center, Oklahoma City, OK 73104, USA; 5Center for Applied Neuroscience, Las Vegas, NV 89101, USA

**Keywords:** concussion, ImPACT, network analysis, neuropsychology

## Abstract

Objective: High school athletes are administered ImPACT at the start of the academic year or sport season and again after suspected concussion. Concussion management involves the comparison of baseline and post-injury cognitive scores with declines in scores providing evidence for concussive injury. A network framework may provide additional information about post-concussive cognitive changes and expand characterization of sport-related concussion (SRC) recovery. Design: Retrospective cohort study. Setting: High school. Participants: High school athletes (*n* = 1553) were administered ImPACT at baseline (T1), post-SRC (T2 = 72 h of injury), and prior to return to play (T3 = within two weeks post-injury). Independent Variables: ImPACT cognitive subtest scores. Main Outcome Measures: Cognitive networks were calculated and compared over three time points. Centrality indices were calculated to determine the relative importance of cognitive variables within networks. Results: Network connectivity increased from T1 to T2 and remained hyperconnected at T3. There was evidence of network reorganization between T1 and T3. Processing speed was central within each network, and visual memory and impulsivity became more central over time. Conclusions: The results suggest potential evidence of cognitive network change over time. Centrality findings suggest research specific to visual memory and impulse control difficulties during the post-concussion recovery period is warranted. Network analysis may provide additional information about cognitive recovery following SRC and could potentially serve as an effective means of monitoring persisting cognitive symptoms after concussion.

## 1. Introduction

Sport-related concussion (SRC) is considered a public health concern among high school, collegiate, and professional athletes [1]. Physiological damage from SRC occurs in stages, starting with initial impact or change in velocity followed by a cascade of neurometabolic changes (e.g., bioenergetic challenges, potential cytoskeletal and axonal alterations, disrupted neurotransmission) [2]. SRC disrupts a variety of cognitive abilities, including attention, memory, processing speed, cognitive flexibility, and other aspects of executive functioning [2,3]. Sleep disturbance, dizziness, and emotional and behavioral changes are also common following SRC [1].

Post-concussive symptoms and cognitive changes typically improve within the first two weeks post-injury and resolve by four weeks post-injury [1,4,5]. On average, athletes return to play (RTP) after two weeks following SRC, although several factors (e.g., initial post-injury symptom burden, type of sport activity) can influence the RTP timeline [4]. Neurocognitive testing is routinely implemented to assist in SRC management and inform RTP decisions. Many athletic programs use Immediate Post-Concussion Assessment and Cognitive Testing (ImPACT) to track symptom severity and cognitive disruption following SRC [6]. ImPACT generates a total symptom score and five cognitive composite scores that represent different domains of functioning. Baseline and post-SRC scores are compared, and declines in composite scores following injury point to SRC-related difficulties [6].

The method of comparing composite scores fits under a “common cause” framework, which assumes that association among variables can be accounted for by a single construct. Competing models, such as network analysis, offer additional information about cognitive functioning. Network frameworks assume cognitive performance is a byproduct of dynamic interactions among different cognitive processes [7]. Thus, growth or decline in one aspect of cognitive functioning is partly independent and partly based on that of other cognitive abilities. Network models can provide nuanced and descriptive information about cognitive functioning while maintaining that abilities are interrelated [8].

Network analysis, a psychometric method for studying associations among variables, has the potential to characterize SRC recovery while considering interconnectedness and mutual facilitation of cognitive abilities. Researchers have investigated the association among self-reported post-concussion symptoms using network analysis. Results highlighted key symptom targets for intervention and revealed idiosyncratic features of post-concussive symptomatology [9,10,11,12,13,14]. A network perspective may be equally illuminating for neurocognitive performance following SRC. Such models can characterize how cognitive abilities influence each other over time [7] and identify cognitive deficits that may be responsible for persisting difficulties.

The present study leveraged a large retrospective cohort of high school athletes to (1) model cognitive recovery following suspected SRC using network analysis, (2) examine whether ImPACT cognitive networks significantly change over time, and (3) identify the relative importance of individual subtest scores. Research on healthy children and adolescents shows that the cognitive network structure remains stable over time [15,16]. Thus, it is possible that an intervening event (i.e., suspected SRC) may lead to changes in network connectivity and structure.

## 2. Methods

### 2.1. Participants

Participants (*n* = 1553) were selected from a larger longitudinal, state-wide, naturalistic database of 80,436 high school athletes who were administered ImPACT (Version 2.0 and Version 2.1) as part of routine concussion management within their respective athletic programs. Athletes were selected if they had three or more consecutive assessments: baseline assessment (T1), post-SRC assessment within 72 h following suspected SRC (T2), and at least one follow-up post-SRC assessment (T3). Athletes’ final assessment was retained if they had several follow-up post-injury assessments as this best represented when athletes returned to play. There was an average of 286.3 days between baseline and acute assessments and an average of 13.38 days between acute and recovery assessments. Athletes did not experience severe injury characterized by loss of consciousness greater than 30 min or post-traumatic amnesia greater than 12 h. Participants were excluded if they completed a post-concussion assessment prior to their first full concussion assessment sequence (i.e., three or more consecutive assessments) to help control for the effects of multiple concussions. Eligible athletes also met the following criteria: ImPACT was administered in English; no self-reported neurodevelopmental disorders, history of brain surgery, or treatment for epilepsy, migraine, substance use, or psychiatric illness (Appendix A). Demographic information is presented in Table 1.

### 2.2. Measure

ImPACT is a multi-domain, computerized neurocognitive test that is widely used for documenting baseline functioning, characterizing effects of SRC, and monitoring recovery [6]. ImPACT includes a demographic survey, a brief medical history questionnaire, a post-concussion symptom scale (PCSS) and six neurocognitive tests (Appendix A) that produce individual scores and five composite scores (Appendix A). The PCSS measures severity of 22 commonly reported post-concussive symptoms. Each item is rated on a 7-point intensity scale ranging from 0 = Not experiencing this symptom, 1 = Barely noticeable, 6 = Worst I have experienced. Items are summed to create a total symptom score, with higher scores representing more severe concussion symptoms. ImPACT software automatically calculates the following five composite scores: Verbal Memory Composite, Visual Memory Composite, Visual Motor Speed Composite, Reaction Time Composite, Impulse Control Composite. The Verbal Memory Composite is calculated by taking the average of the following subtest scores: total percent correct from the Word Memory subtest, total correct items on the hidden trial of the Symbol Match subtest, and the total percent correct from the Three Letters subtest. The Visual Memory Composite is calculated by taking the average of the following subtest scores: total percent correct on the Design Memory subtest and total correct on the memory trial of the X’s and O’s subtest. The Visual Motor Speed Composite is calculated by taking the average of the following subtest scores: total correct on the interference trial of the X’s and O’s subtest and average counted correctly on the interference trial of the Three Letters subtest. The Reaction Time Composite is calculated by taking the average of the following subtest scores: average correct reaction time on the interference task from the X’s and O’s subtest, average correct reaction time on the visible trial from the Symbol Match subtest, and the average correct reaction time on the Color Match subtest. The Impulse Control Composite is calculated by taking the sum of the following scores: total incorrect on the interference phase of the X’s and O’s subtest and total commission errors on the Color Match subtest. ImPACT has adequate psychometric properties, including good construct validity [17] and test–retest reliability [18,19,20]. ImPACT has been demonstrated to increase the sensitivity of post-concussion assessment beyond symptomatic evaluation and physical examination [21,22].

### 2.3. Procedure

ImPACT was administered by trained school personnel within the athletes’ high schools. Baseline assessments were conducted in groups prior to starting the academic year or sport season. Acute post-concussion assessments were conducted on an individual basis within 72 h of suspected concussion. Probable concussions were identified by athletic training staff after injuries were sustained during competitive play and practice. Follow-up post-SRC assessments were administered within one to two weeks following injury if the athlete continued to report symptoms. ImPACT software generates an alternate form with each administration to reduce practice effects [6]. Athletes completed their assessment in a quiet testing environment while supervised by a properly trained healthcare provider. Test administration was completed during a single session and lasted 20 min.

### 2.4. Analyses

#### 2.4.1. Network Architecture

Network analyses were conducted in R version 4.0.3. using qgraph [23], bootnet [24], networktools [25], and NetworkComparisonTest [26]. Network analysis allows for the graphical representation of observed clinical elements (nodes) and the statistical relationship among them (edges). Edge-weight, or the strength of the statistical relationship, is represented by line thickness, while the direction of the statistical relationship is represented with line color [27].

Networks were estimated from unique (i.e., not aggregate, inverse, or derived) ImPACT neurocognitive subtest scores at each time point to yield three separate networks. Given that the data were not normally distributed, a transformation was applied to the data before estimating the networks. The nonparanormal SKEPTIC transformation is so named because the “Spearman/Kendall estimates” preempt transformations to infer correlation [28]. Networks were estimated such that edges represented partial correlations between variables after adjusting for the influence of all other variables in the network. Networks were regularized using the recommended graphical least absolute shrinkage and selection operator (LASSO) algorithm [29]. This process removes weak and spurious edges and returns a sparse network in which a small number of likely genuine edges are used to explain network structure [24]. Graphical LASSO uses a tuning parameter to control the degree of regularization. The higher the tuning parameter, the greater likelihood that spurious edges are dropped from the network [24]. The tuning parameter was selected by minimizing the Extended Bayesian Information Criterion (EBIC), which works well to retrieve the true network structure [24,30]. Hyperparameter γ was set to 0.5, which is recommended for estimating parsimonious networks and balancing sensitivity and specificity [31].

#### 2.4.2. Centrality Analysis

Centrality indices were computed to determine cognitive nodes’ relative importance within the network. Node strength and expected influence represent how strongly a given cognitive ability is directly connected to other cognitive abilities in its network. Strength takes the sum of the absolute edge-weights, while expected influence considers both positive and negative edges [27,32]. Higher values indicate greater node importance [27,32].

#### 2.4.3. Network Accuracy

Three aspects of network accuracy were tested: (1) edge-weight accuracy, (2) centrality stability, and (3) edge-weight and centrality differences [26]. To assess edge-weight accuracy (i.e., the accuracy of estimated network connections), nonparametric bootstrapped confidence intervals (CIs, 95%) were constructed around the regularized edge-weights. Large CIs suggested that edge-weights did not significantly differ. A case-dropping subset bootstrap approach was employed to assess centrality stability. The CS-coefficient signifies the maximum proportion of cases that can be dropped while maintaining a large correlation (*r* = 0.70) between the full- and subset-sample networks’ centrality values. CS-coefficients should be above 0.50 and no lower than 0.25 for the centrality indices to be trustworthy [27].

#### 2.4.4. Sensitivity Analysis

Edge-weight and node centrality differences were examined using calculated difference scores for each pair of bootstrapped edge-weights/centralities. Edge-weights and centralities are likely trustworthy if zero is included in the bootstrapped CI. Goldbricker analysis [25] identified overlapping nodes by calculating the proportion of correlations that were significantly different for each pair of nodes. Nodes with topological overlap greater than 25% (α = 0.01) were considered colinear [25].

#### 2.4.5. Network Comparison

Networks were compared to determine whether there were significant changes across time (T1–T2, T1–T3, T2–T3). No method currently exists to compare the stability of more than two networks at a time [27]. Permutation tests (based on 1000 iterations) assessed differences between two networks based on global strength invariance (i.e., whether overall connectivity was equal across networks) and network structure invariance (i.e., whether network structure and edge-weight distribution were equal across networks) [26,33]. Centrality differences were examined to determine whether influential nodes changed over time. Network layouts were constrained to be equal for all three networks to aid in interpretation. False Discovery Rate (FDR = 0.05) correction was applied to account for multiple edge comparisons [34].

#### 2.4.6. Supplemental Analyses

As a separate set of analyses, a two-factor repeated-measures analysis of variance (ANOVA) was conducted to examine differences in cognitive composite scores over time. Time (T1, T2, T3) and composite domain (Verbal Memory, Visual Memory, Visual Motor Speed, Reaction Time, and Impulse Control) were entered as the two within-subjects factors. Post hoc dependent-samples *t*-tests with Bonferroni correction (*p* = 0.05/3, T1–T2, T1–T3, T2–T3) were conducted for each composite domain if there was a significant interaction to examine direction of effects (i.e., whether scores improved or declined over time). Data were centered prior to analysis, and Reaction Time and Impulse Control were reverse scored for consistency in direction of effects.

## 3. Ethical Considerations

This study used de-identified archival data, which were deemed exempt by the local social/behavioral institutional review board for the protection of human subjects.

## 4. Results

### 4.1. Network Architecture

Out of a possible 276 edges, 197 (71%) were retained at T1, 183 (66%) were retained at T2, and 192 (69%) were retained at T3 (Figure 1). Networks consisted of both positive and negative edges. Across all networks, the strongest edges linked nodes within the same cognitive subtest. Strong edges were similar across networks, but edge 11–13 emerged as a strong edge in the T2 and T3 networks.

### 4.2. Centrality Results

In the T1 network, node 11 (XO average correct reaction time interference) was highest in strength centrality and node 6 (design memory correct distractors) was highest in expected influence. In the T2 network, node 10 (XO total correct interference) was highest in strength and node 6 was highest in expected influence. In the T3 network, node 11 was highest in strength and node 21 (color match average commissions reaction time) was highest in expected influence (Appendix A). At each time point, strength and expected influence (CS (cor = 0.70) = 0.75) were stable under subsetting cases and therefore interpretable (Appendix A).

### 4.3. Network Accuracy Results

CIs were tight around the parameter estimates for edge-weight, suggesting accurate estimation of edge-weight values (Appendix A). Although weaker edge-weight CIs overlapped, there was no overlap around the strongest edges in the networks, suggesting the order of strongest edges are interpretable.

### 4.4. Sensitivity Analysis Results

There were significant differences among edges and centrality indices within each network (Appendix A). Goldbricker analyses identified potential colinear nodes in each of the networks. Node 5 (design memory hits immediate) and node 7 (design memory hits delay) shared topological overlap in each network, and node 2 (word memory correct distractors immediate) and node 4 (word memory correct distractors delay) shared topological overlap in the T2 and T3 networks. Other problematic node pairs included node 6 (design memory correct distractors immediate) and node 8 (design memory correct distractors delay), node 20 (color match total commissions) and node 21 (color match average commissions reaction time), node 11 (XO average correct reaction time) and node 21 (color match average correct reaction time), and node 1 (word memory hits immediate) and node 3 (word memory hits delay). Potential solutions include statistically combining the nodes or excluding one of the nodes in the pair [25]. While these nodes are similar, they tap unique cognitive processes, so excluding one of the nodes or combining scores would result in a loss of important clinical information that is relevant to this population. Thus, all nodes were retained in the networks, and centrality results were interpreted with caution. In sum, all three estimated network structures were accurate, stable, and interpretable.

### 4.5. Network Comparison Results

Network connectivity was significantly greater for T2 compared to T1 (*S* = 2.29, *p* = 0.02), where *S* equals the global strength difference of the two tests [26]. There was no significant difference in network structure between T1–T2 (*M* = 0.56, *p* = 0.08), where *M* equals the maximum difference in any of the edge-weights [26]. T2 and T3 networks did not significantly differ in overall connectivity (*S* = 0.56, *p* = 0.82) or structure (*M* = 0.50, *p* = 0.14). Network connectivity was significantly greater for T3 compared to T1 (*S* = 1.73, *p* = 0.04), and network structure significantly differed (*M* = 0.25, *p* = 0.03). Several nodes significantly differed in strength and expected influence over time (Appendix A). Overall, nodes measuring visual memory became more central and influential over time, and nodes measuring impulsivity became more central from T2–T3.

### 4.6. Supplemental Cognitive Composite Score Comparisons

Repeated-measures ANOVA revealed a significant interaction (*F* (8,12,416) = 55.18, *p* < 0.001, *MSE* = 0.43, η_p_^2^ = 0.03), indicating scores change differently over time (Figure 1). Post hoc *t*-tests revealed a consistent pattern within each cognitive domain in which composite scores significantly worsened from T1–T2 and significantly improved from T2–T3 (Appendix A).

## 5. Discussion

The present study utilized network analysis to assess cognitive performance following suspected SRC. The results revealed network changes over time. T3 exhibited increased network connectivity compared to T1, a pattern that emerged between T1-T2, suggesting the network at two weeks post-SRC remained more similar to the acute post-SRC network than the baseline network. Moreover, network structure significantly changed from T1-T3, reflecting a reorganization of connections among nodes.

Increased connectivity in the ImPACT network may parallel functional brain network hyperconnectivity following SRC. Research suggests that athletes with SRC exhibit functional network hyperconnectivity both acutely and subacutely after injury at the whole-brain level and within key networks [35,36,37,38]. Brain network hyperconnectivity appears to represent compensatory recruitment of additional resources following network disruption [37,38]. Similarly, increased connectivity in the ImPACT network could possibly represent increased effort and engagement of more cognitive resources to meet task demands following injury. Furthermore, network reorganization at two weeks post-injury may reflect use of compensatory strategies to achieve pre-injury cognitive performance. Interestingly, symptom scores and cognitive composite scores seem to normalize at T3, which suggests clinical improvement. It is possible that network models of cognitive recovery can detect nuanced changes that would otherwise be missed when using domain-level scores. Future research is needed to determine the clinical implications of cognitive network hyperconnectivity.

Processing speed, visual memory, and impulsivity nodes were responsible for network maintenance over time. Processing speed nodes were consistently high in strength centrality within each network, suggesting performance on these measures was highly associated with performance on other cognitive tasks. Consequently, processing speed deficits would likely contribute to poor performance on other tasks. Indeed, slowed processing speed after brain injury contributes to self-reported memory problems, attentional deficits, poor concentration, heightened distractibility, and cognitive fatigue [39]. Visual memory nodes exhibited increased strength centrality and expected influence over time suggesting that difficulties with visual memory tasks may impede performance on other cognitive tasks. Similarly, impulsivity nodes became more central after concussion, which highlights their relative importance during the two-week recovery period. Perhaps as time progresses after injury, deficits in impulse control become more dominant in the cognitive network and contribute to disruption on other tasks.

Overall, these central nodes appear to reflect cognitive skills that influence performance on other cognitive tasks. It is possible that improved performance on these domains would help performance on other, more peripheral abilities represented in the network. Given their central roles in the networks, visual memory and impulse control could be the focus of future research on intervention development. Research on the use of cognitive interventions for SRC recovery is nascent [40,41]; however, evidence from the traumatic brain injury literature suggests that compensatory memory aids [42] and mindfulness-based meditation [43] are useful for managing visual memory and impulse control difficulties, respectively. Future research should examine whether rehabilitation treatments specific to these domains are effective for athletes with SRC.

Generalizability of the current results may be limited by the exclusion of athletes with histories of psychiatric, neurodevelopmental, and other disorders that could impact cognitive networks. As a first step, these variables were controlled in the current study to better determine fundamental processes involved in recovery post-SRC. Future research could validate this work within a more diverse sample and examine whether premorbid characteristics influence network structure, stability, and node importance. Furthermore, our sample is predominantly male (63.7%), which may bias the results. Sex differences on ImPACT cognitive performance are mixed, with females outperforming males on some subtests and males outperforming females on others [44]. Future work should examine whether sex influences baseline ImPACT networks or changes in network structure following SRC.

Moreover, while ImPACT is widely used in concussion management, a more comprehensive battery of neuropsychological measures would allow for thorough assessment of cognitive networks in athletes with SRC. Additionally, it is unclear whether network changes observed in this study are solely attributable to effects of SRC. While the repeated-measures design allows for each athlete to serve as their own control, inclusion of an uninjured comparison group measured at similar intervals would help control for normal variation in test performance over time. That said, cognitive networks have been shown to remain stable throughout childhood and adolescence [15,16], and research suggests that athletes perform similarly on ImPACT baseline tests over a two-year period [19].

There are limitations to the immediate clinical application of these findings. For instance, concussions are only “suspected” in this naturalistic sample because there may have been variability in diagnostic criteria used across athletic programs. It is also possible that there was variability in test administration along with equipment computing differences across sites. This work should be replicated in a controlled laboratory setting to account for possible measurement error.

## 6. Conclusions

This study provides potential evidence of changes to the ImPACT cognitive network structure following suspected SRC. Examining node centrality within and across networks identified influential cognitive abilities that may impact performance on other tasks. The current model has the potential to inform the development of evidence-based and targeted treatment approaches for SRC. Future research is warranted to determine the clinical utility of network analysis in concussion management.

## Figures and Tables

**Figure 1 brainsci-13-00177-f001:**
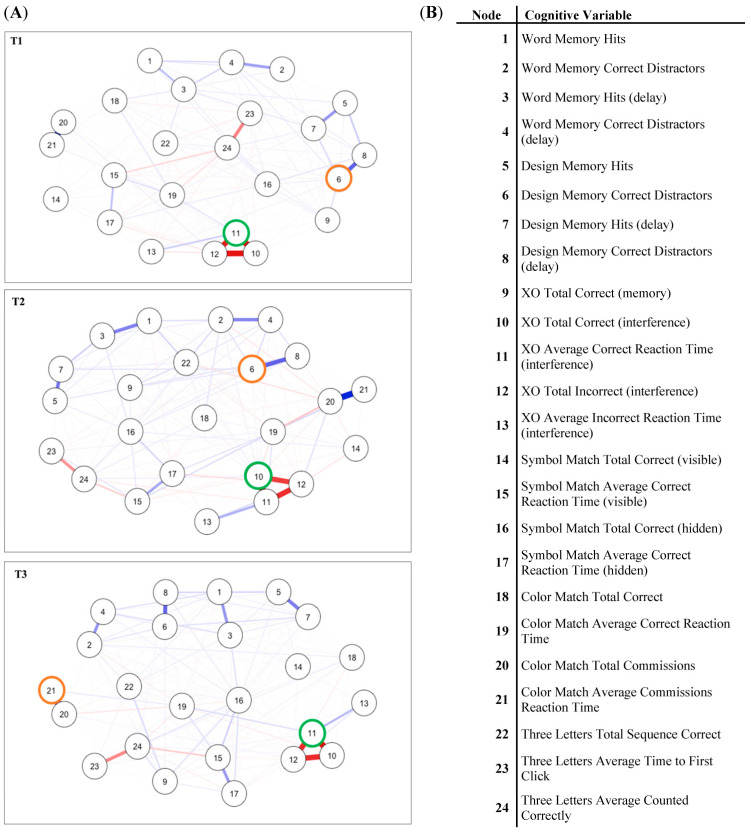
ImPACT Cognitive Subtest Networks and Node Labels. Note. (**A**) T1 (top), T2 (middle), and T3 (bottom) networks with unconstrained layout. (T1 = baseline assessment, T2 = initial administration of ImPACT within 72 h of suspected concussion, T3 = second administration of ImPACT within 14 days of suspected concussion). Highest node strength outlined in green, highest expected influence outlined in orange. (**B**) Node numbers and corresponding variable names for the 24 cognitive variables.

**Table 1 brainsci-13-00177-t001:** Demographic Information by time point (*n* = 1553).

	T1	T2	T3
Age			
Mean (SD)	14.7 (1.02)	15.5 (1.16)	15.5 (1.16)
Sex			
Female Total (%)	564 (36.3%)	—	—
Male	989 (63.7%)	—	—
Symptom Score			
Mean (SD)	5.25 (1.02)	18.7 (1.16)	1.90 (1.16)

Note. SD = standard deviation. Symptom score refers to the total score from the post-concussion symptom scale (PCSS). PCSS items are summed to create a total symptom score, with higher scores representing more severe concussion symptoms. Symptom scores significantly increased from T1 to T2 and significantly decreased from T2 to T3 after *p* < 0.017 (0.05/3 Bonferroni correction). Sex remained the same from T1–T3 (T1 = baseline assessment, T2 = initial administration of ImPACT within 72 h of suspected concussion, T3 = second administration of ImPACT within 14 days of suspected concussion).

## Data Availability

Restrictions apply to the availability of these data. Data requests may be made to T.K. and S.R. http://www.centerforappliedneuroscience.com/ (accessed on 19 January 2023).

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
