# Peer review of "Changes in ImPACT Cognitive Subtest Networks Following Sport-Related Concussion"

_brainsci, 2023, doi:10.3390/brainsci13020177_

Round 1
Reviewer 1 Report
The manuscript describes a revaluation of high-school athlete data from baseline and post sports related concussion at two time-points using a network analysis approach. I greatly commend the systems approach of creating networks of outputs to provide a wholistic measure of cognition.
I have some comments that would need clarifying before recommending for publication.
Table one shows ‘symptom scores’ at T1 5.25 (1.02) T2 18.7 (1.16) and T3 1.90 (1.16)
It is not clear from the manuscript how these scores are arrived at. In the methods the authors state a that the ‘post-concussion symptom scale, and six neurocognitive tests (Supplementary Tables 1 114 and 2) that produce individual scores and five composite scores.’ I see the individual scores in the supplemental table, but no composite score, and it is not clear from the individual scores how T1, T2 and T3 are calculated.
Please can the authors make this clear.
Can the authors please clarify, on average, how long T3 is after suspected concussion? This could have an impact in the interpretation of the data.
Also, more simplistically, I assume, the numbers in table 1 (T1 vs T2) and (T2 vs T3) are significantly different from one another? i.e. basline (T1), cognitive score after SRC (T2) then during recovery (T3). Can the authors statistically compare these?
If yes, does the conclusion of the paper, that the cognitive network starts to become hyperconnected at T2, which continues at T3 (reflecting brain changes), go against the standard cognitive scoring, which recovers (as I assume, so do the concussed individuals clinically)?
Is the paper suggesting that there is a clinical recovery from SRC, reflected by the symptom score, but using the network analysis, we see underlying cognitive changes/strategies that have previously been overlooked?
If yes, this should be stated more clearly.
Author Response
We thank the reviewer for taking the time to review our manuscript. We hope that we have adequately addressed initial concerns and that the manuscript will be suitable for publication in Brain Sciences. We have included a detailed item-by-item response below.
Reviewer 1:
The manuscript describes a revaluation of high-school athlete data from baseline and post sports related concussion at two time-points using a network analysis approach. I greatly commend the systems approach of creating networks of outputs to provide a wholistic measure of cognition.
I have some comments that would need clarifying before recommending for publication.
Table one shows ‘symptom scores’ at T1 5.25 (1.02) T2 18.7 (1.16) and T3 1.90 (1.16)
It is not clear from the manuscript how these scores are arrived at. In the methods the authors state a that the ‘post-concussion symptom scale, and six neurocognitive tests (Supplementary Tables 1 114 and 2) that produce individual scores and five composite scores.’ I see the individual scores in the supplemental table, but no composite score, and it is not clear from the individual scores how T1, T2 and T3 are calculated.
Please can the authors make this clear.
Response: We thank the reviewer for raising this important point. We now describe how each of the five composite scores are calculated (pg. 7-8). We have also added a table to the supplementary materials that shows how each of the five composite scores changed over time (Supplementary Table 3). We also now include how the post-concussion symptom scores are calculated (pg. 18, pg. 28). Given that the PCSS was not our main outcome measure, we felt it was appropriate to place this information in a footnote. We are willing to move this section to the main text if the reviewer feels that is more appropriate.
Can the authors please clarify, on average, how long T3 is after suspected concussion? This could have an impact in the interpretation of the data.
Response: We thank the reviewer for mentioning this point. We do not have the exact date of injury, but given the assessment procedures followed by each athletic program, we were able to estimate that T3 was within 13-16 days of the initial suspected injury. On page 7 of the manuscript, we mention that there are 13.38 days between the acute assessment (T2) and the recovery assessment (T3). We also mention on page 8 that acute assessments were administered within 72-hours of suspected concussion. Therefore, the T3 assessment was likely 14-16 days following the suspected injury. This measurement interval is typical in concussion management. Please let us know if we can clarify further.
Also, more simplistically, I assume, the numbers in table 1 (T1 vs T2) and (T2 vs T3) are significantly different from one another? i.e. basline (T1), cognitive score after SRC (T2) then during recovery (T3). Can the authors statistically compare these?
Response: The reviewer raises an interesting point about comparing the cognitive subtest scores. We agree that it would be helpful to statistically compare subtest scores across time points. However, we hesitate to run that many statistical tests given the high likelihood of inflating Type I error (24 separate comparisons + 3 post hoc tests per comparison). We agree that this is an important question to address, so we statistically compared the five composite scores across each time point using ANOVA. We used a .05/3 Bonferroni correction to correct for multiple comparisons. We now describe these methods and results in the manuscript (pg. 11, 14) and include a table of results in the supplementary materials (Supplementary Table 3). We also statistically compared total symptom scores across time using ANOVA (.05/3 Bonferroni correction), and these results are now listed under Table 1 (pg. 18).
If yes, does the conclusion of the paper, that the cognitive network starts to become hyperconnected at T2, which continues at T3 (reflecting brain changes), go against the standard cognitive scoring, which recovers (as I assume, so do the concussed individuals clinically)?
Is the paper suggesting that there is a clinical recovery from SRC, reflected by the symptom score, but using the network analysis, we see underlying cognitive changes/strategies that have previously been overlooked?
If yes, this should be stated more clearly.
Response: The reviewer makes an interesting point about our findings. Indeed, the cognitive scores and symptom scores seem to improve by T3, while the cognitive network becomes hyperconnected at T2 and persists at T3. It is possible that while domain-level scores suggest clinical recovery, network models suggest cognitive changes. We now state these findings and conclusions in the discussion (pg. 15). We use tentative language here because clinical implications of cognitive networks require more research.
Reviewer 2 Report
The investigate the effect of sports-related concussion (SRC) on ImPACT cognitive subtesting over time in high school athletes. They find that cognitive network connectivity appears to change over time. The findings are important in understanding the consequences of SRC. The manuscript is well-written and scientifically sound. A few changes would improve the manuscript, as follows:
1. How were athletes with multiple SRC handled? This cohort may differ from those with single concussion.
2. A specific conclusion section (with its own heading) would be helpful.
Author Response
We thank the reviewer for taking the time to review our manuscript. We hope that we have adequately addressed initial concerns and that the manuscript will be suitable for publication in Brain Sciences. We have included a detailed item-by-item response below.
Brain Sciences Review Comments
Reviewer 2:
The investigate the effect of sports-related concussion (SRC) on ImPACT cognitive subtesting over time in high school athletes. They find that cognitive network connectivity appears to change over time. The findings are important in understanding the consequences of SRC. The manuscript is well-written and scientifically sound. A few changes would improve the manuscript, as follows:
- How were athletes with multiple SRC handled? This cohort may differ from those with single concussion.
Response: We thank the reviewer for raising this important point. We selected data from participants’ first full concussion assessment sequence (i.e., baseline assessment, acute assessment, recovery assessment). In other words, we excluded participants who had a previous post-concussion assessment prior to this first full concussion sequence. Given that concussions were only suspected in this naturalistic sample, we are unable to confirm whether participants received a concussion diagnosis prior to T1. We mentioned this procedure in our Participant Selection Diagram (supplementary figure 1, pg. 3 of supplementary materials), but we agree that it should be mentioned in the main text. We now include this information on page 7.
- A specific conclusion section (with its own heading) would be helpful.
Response: We agree that adding a conclusion heading would be helpful. We have now included a conclusion section on pg. 17.